# Predictive Dynamics Improve Noise Robustness in a Deep Network Model of the Human Auditory System

## Abstract

Sensory systems are robust to many types of corrupting noise. However, the neural mechanisms that drive robustness are unclear. Empirical evidence suggests that top-down predictions are important for processing noisy stimuli, and the substantial feedback connections in primate sensory cortices have been proposed to facilitate these predictions. Here, we implement predictive dynamics in a large scale model of the human auditory system. Specifically, we augment a feedforward deep neural network trained on noisy speech classification with a recently introduced predictive feedback scheme. We find that predictive dynamics improve speech identification across several types of corrupting noise. These performance gains were associated with denoising of network representations and alterations in layer dimensionality. Finally, we find that the model captures brain data outside of the speech domain. Overall, this work demonstrates that predictive dynamics are a candidate mechanism for human auditory robustness and provides a testbed for hypotheses regarding the dynamics of auditory representations. Additionally, we discuss the potential for this framework to provide insight into robustness mechanisms across sensory modalities.

## 1 Introduction

Our ability to process complex sensory information requires the isolation of stimuli of interest from background noise. For instance, we can pick out a friend's voice in a crowded coffee shop or spot a predator in a grassy field. To distinguish these stimuli of interest, your sensory system must somehow become robust to background noise. A long standing goal in the field of sensory neuroscience is to understand the neural mechanisms that enable human sensory systems to solve these problems [1–3].

A large body of experimental evidence suggests that top-down predictions play an important role in processing stimuli in noisy contexts [4–6]. Feedback connections are abundant in sensory cortices [7, 8], and have been hypothesized to carry predictive signals [9] but are often omitted in large scale models. In the visual neuroscience literature, imbuing computational models with biologically-inspired predictive dynamics has been found to increase performance in noisy object recognition tasks [10–12]. These insights offer a promising indication that predictive dynamics play a role in enabling noise robustness.

Here, we explore noise robustness in the context of auditory perception. We augmented a deep neural network of the human auditory system with predictive dynamics and explored how these dynamics affected network representations and performance. We find that predictive dynamics improve robustness across several types of real-world noise. We explore different mechanisms that support this robustness. Finally, we discuss how the intrinsically temporal nature of auditory

Submitted to 4th Workshop on Shared Visual Representations in Human and Machine Visual Intelligence (SVRHM) at NeurIPS 2022. Do not distribute.

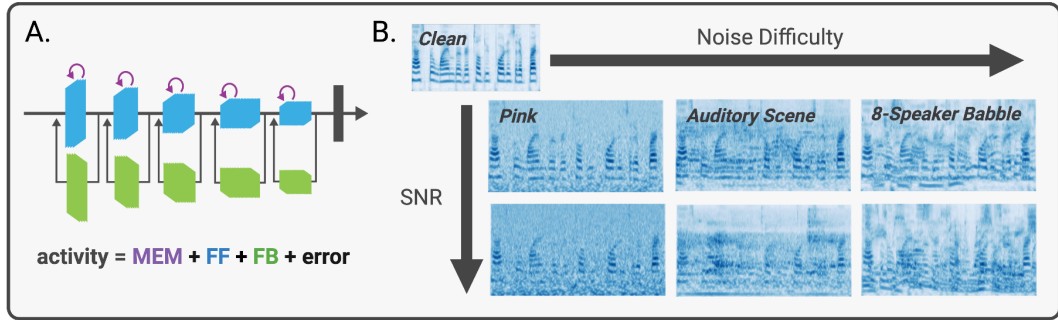

Figure 1: **A.** Schematic of network and predictive dynamics. FF: feedforward, FB: feedback, ER: error correction, MEM: memory. Each block indicates a convolutional layer. Error is not depicted in the diagram. **(B)** Clean sounds were embedded in three background noises of varying difficulty, with different grades of SNR.

perception lends itself to interesting sensory perception problems that are general across many modalities.

## 2 Model

### 2.1 Predictive dynamics in a deep neural network are introduced with the Predify framework.

We use the Predify framework [10] to incorporate predictive dynamics into a feedforward network. Briefly, given input into the network at time 0, the activity of each layer $i$ at some time $t$ is a linear combination of four terms (Fig 1A; see Appendix A.1): **(1)** Feedforward drive taking activity from layer $i - 1$ as input, **(2)** Feedback from layer $i + 1$ that predicts the activity at layer $i$, **(3)** Error correction that computes the gradient of the prediction error of layer $i$ to correct the activity at layer $i$, **(4)** Memory that integrates over the activity from time $t - 1$.

Under this framework, higher-order regions predict and correct the activity of lower-order regions. Each layer has a set of hyperparameters that control the proportional strength of each of these terms. Note that hyperparameters for feedforward, feedback, and memory are constrained to sum to 1. Different settings of the hyperparameters can capture a variety of known dynamics, from purely feedforward to classical predictive coding [10].

### 2.2 Feedforward weights are learned in a supervised auditory classification task

We use the feedforward neural network introduced by [13] to define the feedforward weights of the predictive network. This network was trained to classify words given cochleagram inputs (time-frequency decomposition of sounds modeled after the human cochlea, Appendix A.2). Inputs were embedded in real-world background noise of varying signal-to-noise ratio (Fig 1B).

We evaluate the performance of the network on speech classification with speech embedded in pink noise, auditory scene, and 8-speaker babble (Fig 1B). These backgrounds were chosen because humans show variable speech recognition performance across these conditions ([13]).

This network was chosen because of its behavioral similarity to humans and ability to capture human auditory cortical responses more accurately than other models [13].

### 2.3 Feedback weights are learned in an unsupervised reconstruction task

Consistent with [10], we froze the feedforward weights and trained the feedback connections in an unsupervised fashion. Each feedback layer $i$ was trained to reconstruct the activity at layer $i - 1$ given the activity at layer $i$. Importantly, the inputs used for this training step have no background noise.

Finally, the hyperparameters are optimized as in [12]. Optimization was performed for each combination of background noise and SNR to maximize classification performance on the task from [13].

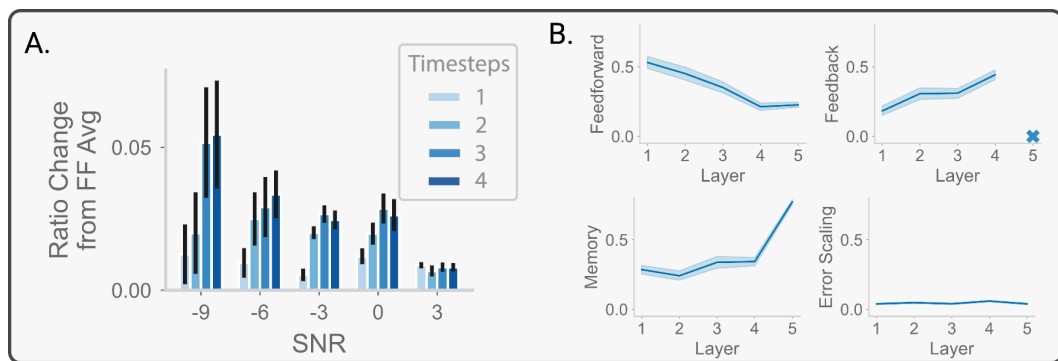

Figure 2: **(A)** Relative increase in performance from the original feedforward on test set for various levels of SNR. 4 timesteps of predictive dynamics were used. SEM is shown for 10 different random initializations of hyperparameter training for each background noise/SNR combination. **(B)** Hyperparameter values of the networks shown in (A). The last (5th) layer of the network does not have feedback by definition.

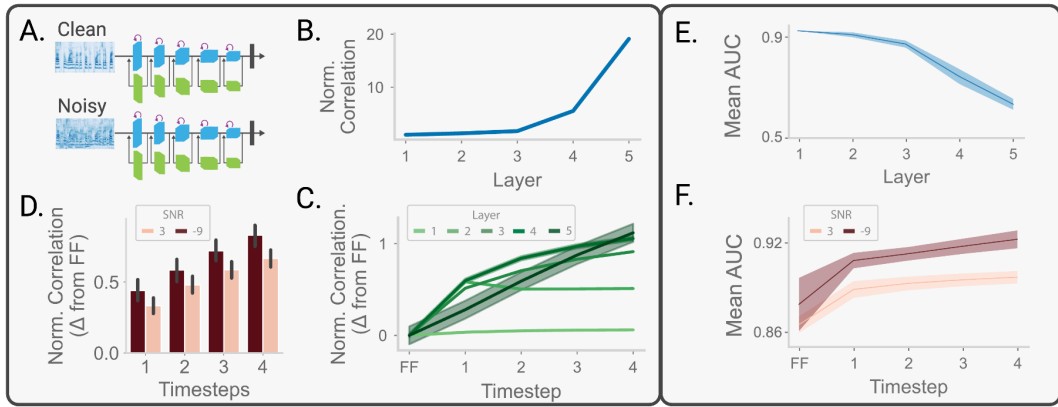

Figure 3: **(A)** Layer representations were compared when the network input was a sound with background noise versus the same sound with no noise. **(B)** Normalized correlation for each layer during the initial feedforward pass. Correlations were normalized at each layer by random shuffles of sound labels. **(C)** Difference in normalized correlation from (B) as predictive time steps are incorporated. **(D)** As in (C), but averaging over all layers and comparing the most noisy (SNR$= -9$) and least noisy (SNR$= 3$) conditions. **(E)** Area under the curve (AUC) of the cumulative singular value spectrum of activity at each layer for feedforward pass alone. **(F)** As in (E), but with predictive timesteps incorporated and comparing the most noisy and least noisy conditions.

## 3    Results

### 3.1    Predictive dynamics improve performance on an auditory classification task.

We found that predictive dynamics improved classification performance relative to the feedforward network alone (Fig 2A). This improvement was more dramatic for more corrupted sounds (i.e. lower SNR) and varied across background noise types (see Appendix A.3).

To better understand the contribution of each model component to this performance improvement, we analyzed fitted hyperparameters across layers (Fig 2B). We found that predictive feedback contributed to dynamics for all layers (i.e. feedback weight $> 0$) and that this contribution increased in deeper layers. Note that a feedforward weight of 1 for all layers is equivalent to the feedforward network. Interestingly, the error correction term (a key element of the canonical predictive coding model [14]) does not play a significant role in network dynamics. In fact, ablating this term had limited impact on the performance of the network (Appendix A.4).

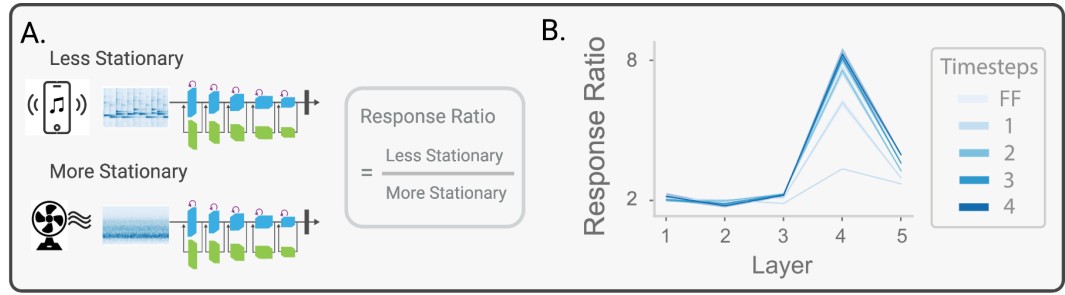

Figure 4: **(A)** Network representations of each layer were compared for less stationary (e.g. ringtone) and more stationary (e.g. air conditioner) sounds. Response ratio was calculated as the mean response to less stationary sounds divided by the response to more stationary sounds. **(B)** Response ratio for each layer of the network, across predictive timesteps.

## 3.2 Noise robustness is associated with denoising of neural representations and altered dimensionality.

Next, we sought to explore the mechanism driving this performance improvement. We hypothesized that predictive dynamics denoise by pushing network representations towards a manifold of clean sounds learned by the feedback weights [10]. To test this hypothesis, we measured network responses to noisy sounds and their clean counterparts. Consistent with the experimental literature [15], we defined denoising as an increase in correlation between responses to clean and noisy sounds across layers (Fig. 3A).

We found that the feedforward network alone shows denoising across layers (Fig. 3B). However, predictive dynamics still appear useful in this denoising process. Across predictive timesteps, there is an additional, albeit smaller, denoising effect (Fig 3C). Consistent with our performance findings, this effect is stronger for inputs that are noisier (Fig 3D).

We also examined the dimensionality of network activity across each layer, a related measure to denoising. Specifically, we computed the cumulative singular value spectrum of the output at each layer. We summarize the dimensionality by computing the area under the curve of this function (AUC). A low AUC corresponds to high dimensionality. We found that the dimensionality of representations increase across layers in the feedforward network alone (Fig 3E). This is consistent with the observation that increasing the dimensionality of representations is useful for classification performance [16]. However, the introduction of predictive dynamics causes dimensionality to decrease with more predictive timesteps, particularly for noisier conditions (Fig 3F). These findings could be consistent with a mechanism by which the initial feedforward weights are trained to maximize separability of representations, predictive dynamics will contract the representations towards a learned clean manifold.

## 3.3 Network activity recapitulates neural data in a non-speech task.

A potentially complementary hypothesis suggests that sensory systems denoise by developing statistical summaries of complex scenes [17–19]. Given that background sounds tend to have more stationary statistics while foreground sounds tend to have less stationary statistics, it has been proposed that the auditory cortex could leverage these statistical differences to distinguish them [20]. A recent fMRI study found support for this hypothesis [13]. Specifically, authors found that early layers of the auditory hierarchy respond similarly to stationary and non-stationary sounds while deeper layers have a higher response ratio of non-stationary to stationary sounds.

We test if our model captures this finding by comparing the responses at each layer to a set of natural sounds with varying stationarity. In the feedforward network, we see a modest increase in response ratio across layers of the network. This effect becomes more dramatic with predictive timesteps (Fig 4B). Consistent with neural data [13], these findings suggest that our model is sensitive to the temporal statistics of natural sounds.

## 4   Discussion

In this work, we sought to evaluate the potential mechanistic role of predictive dynamics in noise robustness using a large scale model of the human auditory system. We found that predictive dynamics improved classification performance across timesteps and that this improvement was most dramatic for more difficult conditions (i.e. lower SNR). The dependence of performance improvement on task difficulty is consistent with other modeling results [10, 12] as well as and empirical findings that that observers rely more on predictions as sensory input becomes less reliable [6]. Additionally, we probed the model by ablation and found that the removal of the error correction term had limited impact on performance. In contrast, the balance of feedforward drive and feedback are essential for performance suggesting more consistency with alternative theories of sensory predictions [21] than canonical predictive coding [14]. This network architecture provides a promising bridge between circuit-level and normative theories of sensory processing.

We probed the mechanism underlying this performance improvement through analyses of network representations. We found evidence that predictive dynamics denoise representations and decrease dimensionality. These findings are consistent with a mechanism by which predictive dynamics denoise by pushing network representations towards a manifold of clean sounds learned by the feedback weights [10]. In future work, we aim to more rigorously evaluate this mechanism through manipulations of the manifold learned by feedback weights.

Finally, we evaluated the extent to which this modeling framework captures findings in literature beyond speech tasks. Specifically, we evaluated model responses to stationary (i.e. background) and non-stationary (i.e. foreground) natural sounds. Consistent with brain data [13, 22, 23], we found that later layers of the model responded preferentially to non-stationary sounds and that this effect became more dramatic across predictive timesteps. Intriguingly, we recapitulate these results despite the fact that the network was only trained on speech and was not exposed during training to other types of natural sounds. This suggests that background invariance could be a general consequence of predictive dynamics.

We acknowledge several limitations on the interpretation of our work. First, training occurred in a step-wise fashion with weights being frozen after each step. While this approach is common in machine learning, it is possible that end-to-end training would yield different solutions. Second, in this predictive coding scheme, uncertainty is not taken into account in the updating process. In Bayes-optimal predictive coding, the integration of sensory evidence with top-down priors is weighted by their respective uncertainty. Allowing for this weighting based for individual stimuli could also yield a different solution. Finally, while we have shown that our model qualitatively captures some findings in the literature, quantitative comparisons between model and human data are needed to demonstrate that this model can provide insight into the human auditory system.

## 5   Broader Implications

Alterations in predictive sensory processing have been widely documented in disease states, such as hallucinations [24–26]. While modeling approaches have provided insight into the cognitive mechanisms that may be disrupted in these states [27, 28], alterations in cortical processing are unclear. Our network model provides a way to probe how circuit-level alterations in predictive dynamics may give rise to pathological states.

We further propose that studies of auditory processing can provide insight into general sensory processing principles. In vision, predictive dynamics are often considered in the context of static images [11, 10, 12]. However, many visual problems require information integration over time. For instance, recognition of a well-camouflaged predator likely requires temporal integration of visual (e.g. motion of predator relative to surroundings). Insights into how the human visual system solves this problem could come from findings in audition, an inherently temporal problem.

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

## A Appendix

### A.1 Predictive dynamics are integrated into feedforward activity over many timesteps

We use the recently introduced Predify framework [10, 12]. A schematic is shown in Fig 5. Under this framework, the network receives input at time $t = 0$. The activity $e$ of layer $i$ at time $t$ is defined as:

$$e_i(t + 1) = \gamma f_i + \beta d_{i+1} - \alpha \nabla \epsilon_i + \mu e_i(t)$$

The feedforward term $f_i = \mathcal{F}_i(e_{i-1}(t + 1); W_i^f)$ is the output of a convolutional layer $\mathcal{F}$ parameterized by weights $W_i^f$. The feedback term $d_{i+1} = \mathcal{D}_{i+1}(e_{i+1}(t); W_{i+1}^b)$ is the output of a deconvolutional layer $\mathcal{D}$ parameterized by weights $W_i^b$. Finally, the error correction term $\nabla \epsilon_i = \nabla_{e_i} MSE(d_i, e_{i-1})$. Each term is weighted by hyperparameters $\gamma, \beta, \alpha, \mu$. Importantly, there is the constraint $\gamma + \beta + \mu = 1$.

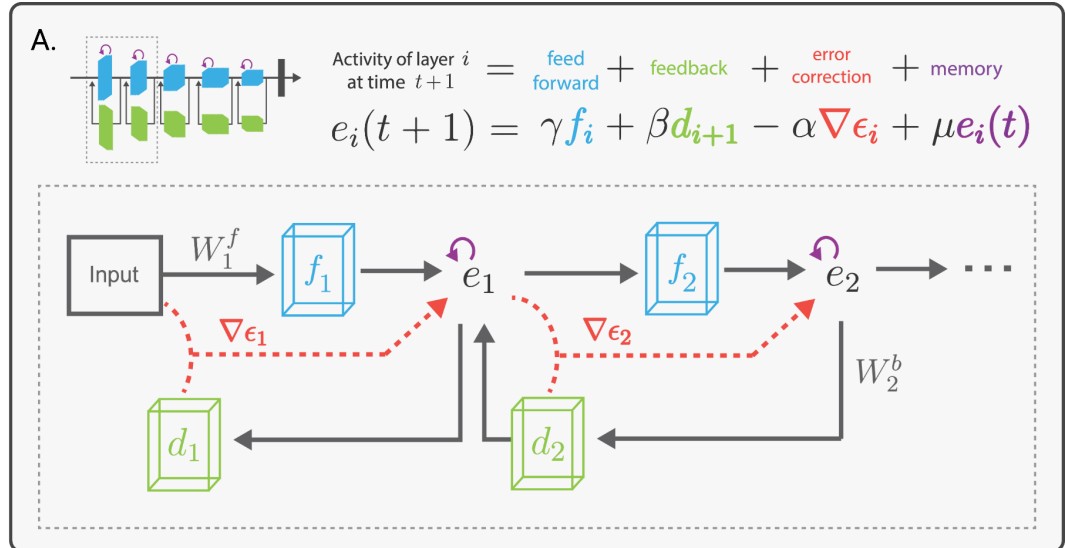

Figure 5: **(A)** Schematic of the first two layers of the network and the predictive dynamics equation. The activity of layer $i$ at time $t + 1$ is denoted as $e_i(t + 1)$ and is computed as a linear combination of feedforward, feedback, error correction, and memory terms. In the schematic, blue blocks represent convolution layers (parameterized by weights $W_i^f$ for layer $i$), while green blocks represent deconvolution layers (parameterized by weights $W_i^b$ for layer $i$).

### A.2 Training details for predictive deep network model of auditory processing

Here we provide more details of the training for the predictive network (Fig 6). Speech for the word classification task was sampled from the Wall Street Journal corpus [29]. The sounds are converted to cochleagrams as in [13] using pyCocleagram (https://github.com/mcdermottLab/pycochleagram). Cochleagrams are time-frequency decomposition of sounds with frequency dependent bandwiths and compressive non-linearities similar to the human cochlea.

Background noise is added as in [13]. Additionally, we introduce pink noise as a new background noise type to establish a particularly easy noise condition with highly stationary statistics. The feedforward network is trained for word classification only on noisy sounds [13]. Feedback weights are trained on an unsupervised reconstruction task with clean sounds [10]. Finally, hyperparameters are learned via backpropagation to maximize performance on word classification on noisy sounds [12]. Importantly, hyperparameters are trained separately for each combination of background noise and SNR level [12]. All analyses are performed on a held-out validation set not seen in any of the training procedures. All error bars and error shading shown are standard error.

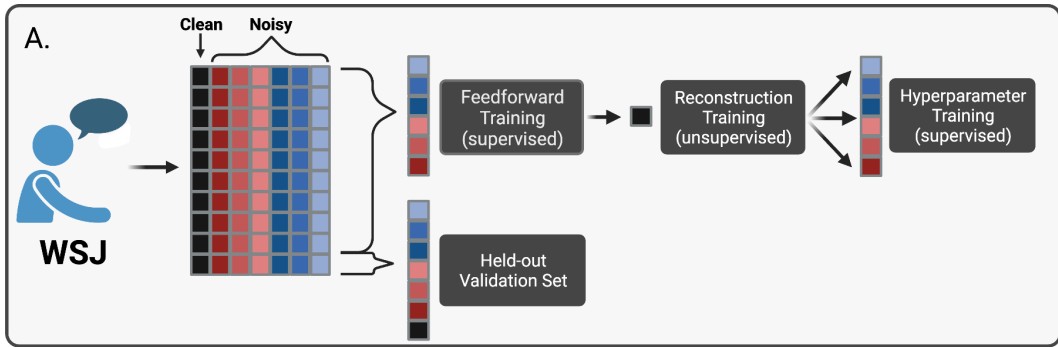

Figure 6: **(A)** Schematic of dataset generation and how it is split amongst the different training procedures. Each square represents some 2-second sound from the WSJ corpus. The color represents the SNR of the sound. Sounds either have no added noise (black) or have added noise of SNR $-9, -6, -3, 0, 3$ decibels (red through blue).

### A.3 Model performance, split across each background type

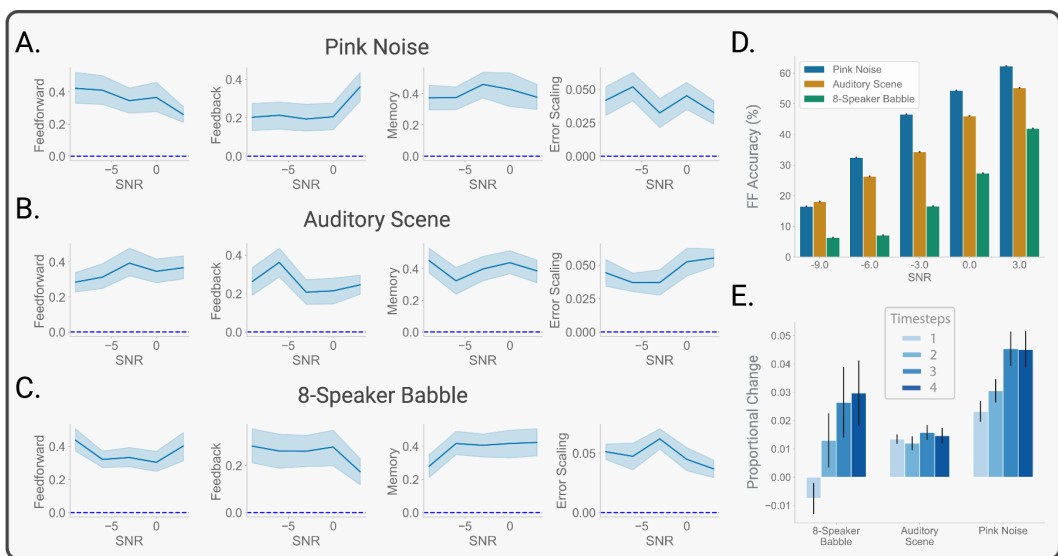

Figure 7: **(A)** Hyperparameter values of the trained networks, but only for the networks trained on pink noise. **(B)** As in (A), but only for auditory scene. **(C)** As in (A), but only for 8-speaker babble. **(D)** Feedforward accuracy on held-out data set, split by background noise. **(E)** Ratio change of accuracy from feedforward across predictive timesteps, evaluated on held-out data set and separated by background type.

### A.4 Model performance with error correction term ablated

To test how crucial the error ablation term is, we set the associated hyperparameter $\alpha$ to 0 during the hyperparameter training and during the network evaluation. We find that the hyperparameter values and performance of the network do not appear to be impacted by this ablation (Fig 8).

### A.5 Potential societal impacts

We do not foresee negative societal impacts from this study, as we are focused on basic neuroscience contributions.

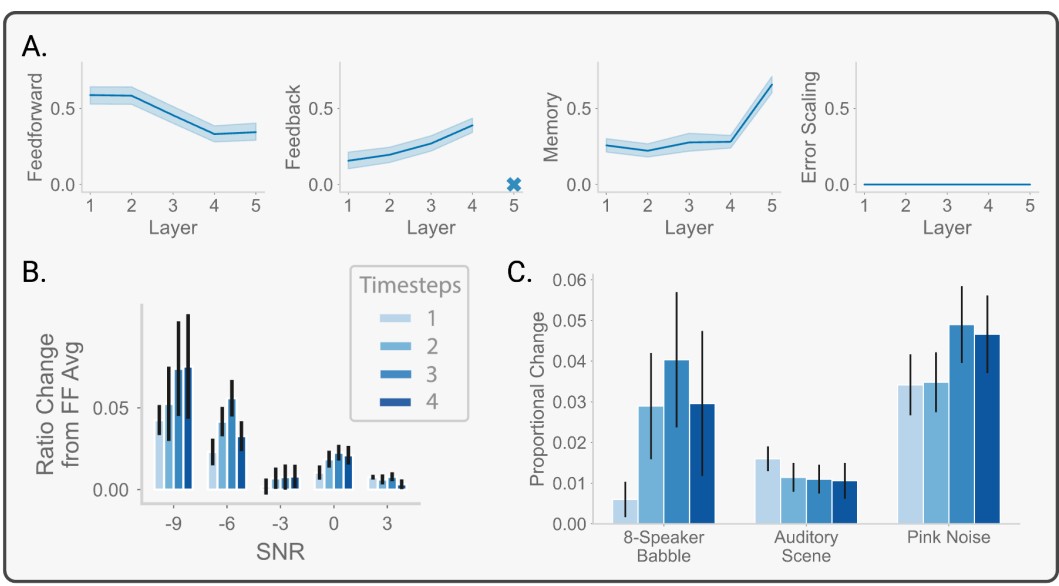

Figure 8: **(A)** As in Fig. 2B but with error correction term ablated. **(B)** As in Fig. 2A, but with error correction term ablated. **(C)** As in Fig. 7E, but with error correction term ablated.

