# OpenReview forum: "Predictive Dynamics Improve Noise Robustness in a Deep Network Model of the Human Auditory System"
_NeurIPS.cc/2022/Workshop/SVRHM — SVRHM Oral_

### Official Review · Reviewer_Fv19 · 2022-10-05
**Clear and relevant paper about the benefit of PC dynamics in noisy auditory tasks.**

**Rating:** 8
**Confidence:** 4

**Review:**

Overall the paper is clear and well structured, and the results are of interest and novel. In the following, I point out some minor comments or things that are not clear from the text and may be added to improve clarity.

- In the model’s equation, the error correction is computed with respect to the layer l-1 or l+1?

- Why the error correction term has a parameter that does not sum up to 1 with the others?

- It is unclear, in the results concerning the hyperparameters as a function of the layer, what was the noise level (the SNR) for the results shown in figure 2B?

- In my understanding, the prediction-error is the one best capturing the essence of PC dynamics. Why do the authors think it’s actually not so relevant in the dynamics (figure 2B)? Could it be that’s because it was not included in the overall dynamics (i.e. it doesn’t sum up to 1 with the other so doesn’t play a role in the push-pull effect ?)

- Figure 3C shows an increase of the « denoising » with deeper layers: could the authors explain or speculate why that would be the case?

-In paragraph 3.2, it could be useful (for a naive reader like this reviewer)  to explain how the dimensionality is related to denoising. Also, the interpretation of the results is not entirely clear: given these results, how can we conclude that FF networks maximize separability and feedback dynamics improve manifold projection?

-Paragraph 3.3: how is performed exactly the comparison between each layer activity and natural sound? How does the ratio between mean responses provide a measure of the learned statistics?

---

### Official Review · Reviewer_GZe4 · 2022-10-13
**An interesting look at the role of predictive dynamics for robust sensory systems**

**Rating:** 7
**Confidence:** 4

**Review:**

Overall, I think this paper should be accepted. The motivation and results are very clear, and the work is a good contribution toward understanding how predictive dynamics may contribute to robustness in sensory systems.

### Summary:
Predicitve dynamics might make human sensory system robust.  The authors probe the relationship between predictive dynamics and robustness using deep neural networks (DNN) train on a speech classification task.   They find that a DNN trained with predictive dynamics (using the Predify framework) are better at identifying speech under noisy conditions than a standard feed-forward network.  The predictive dynamics model also shows denoising properties in its representations and reduced dimensionality.  The authors finally show their model is sensitive to temporal sound statistics by comparing model response to neural recordings.

### Quality/Clarity:
The paper is well written and organized

### Originality:
The application of Predify to speech classification is original and so are the analyses.

### Significance:
I think it is a valuable contribution to have more auditory DNNs with biological mechanisms. Many people are evaluating static vision models, but as the authors point out, audio is an interesting space to test out temporal representations.

### Pros:
- Clear hypothesis and model testing set-up
- Multiple representational analyses

### Cons:
- More comparison to human behavioral baselines would have strengthened the results
- More explanation about neural activity findings would have strengthened the point about the paper’s significance for tasks other than speech classification

---

### Official Review · Reviewer_PRfc · 2022-10-13
**Excellent study of the computational advantages of predictive coding in auditory classification**

**Rating:** 9
**Confidence:** 5

**Review:**

This paper introduces predictive coding dynamics into a convolutional neural network trained for auditory classification. The feedback connections improve accuracy and robustness under various types of noise. Further, the authors explore the underlying reasons for this improvement: predictive coding iterations make the noisy patterns more similar to the corresponding clean patterns, and reduce the intrinsic dimensionality of the data. This strongly supports the notion that predictive feedback could play a key role in sensory processing. The paper is well-written and very clear given the space allotted.

A couple minor issues:
* The final discussion suggests that given the temporal nature of auditory signals, the findings could also have implications for temporal integration in vision. While this might be true in general, in the present case I think this is an overstatement, because the "temporality" of the auditory signals is pre-processed into a 2D spatial signal, which is then treated as a static image. I would be more cautious with this sort of statement.
* The paper by Choksi et al has now been published in Neurips, so the arxiv citation should be updated.

---

### Official Review · Reviewer_y9hR · 2022-10-16
**Solve the cocktail-party problem with auditory top-down prediction**

**Rating:** 8
**Confidence:** 4

**Review:**

The paper investigates how the neural mechanism drives robustness in the human auditory system by utilizing the predictive dynamic in deep neural network and testing it on speech tasks corrupted by different kinds of noise. The results show that predictive dynamics can improve the performance of speech identification. The paper is well written, the architecture is well framed, the experiment is clearly described, and the result is clearly represented. However, the network architecture and optimization technique come from previous work. Furthermore, several figures don't support their conclusions.

Line 71, in Appendix A.3, Figure 7A-C show the values of the four hyperparameters (columns) and three kinds of noise (lines) varied on different SNR. There are no further explanations for those twelve panels. The variation is large, and there is no obvious trend. Should the readers expect to observe higher feedback values for the lower SNR situations?

Line 75-78, the author claimed that the error correction term does not play a significant role in network dynamics. However, Figure 8B is different from Figure 2A. For instance, when SNR is -9dB and Timesteps is 4, the ratio change is 0.07 vs. 0.05. Does this mean that prediction error ablation increases the performance by 40%?

Line 111, should mention Figure 4A here.

Line 111-113, Figure 4B confuses me for two reasons. First, why is there a dramatic increase from layer 3 to layer 4? If this network has 6 layers, will the response ratio still increase dramatically, mildly, or be saturated? And second, why the response ratio decreases at layer 5? In addition, the authors mention that Figure 4B is consistent with neural data. However, in the monkey visual (V1, V2, V4, and IT) and auditory (core, belt, and parabelt) ventral pathways, I didn't see any studies show a dramatic increase then decrease for encoding stimulus features. The change from the thalamus to the cortex may be dramatic, but within the cortex, it should be modest and monotonic.

Line 156-161, check this paper: deep predictive coding networks for video prediction and unsupervised learning.

Line 273, should be the activity e of layer i at time t+1 instead of time t.